



# Precision Annealing Monte Carlo Methods for Statistical Data Assimilation: Metropolis-Hastings Procedures

Adrian S. Wong [1], Kangbo Hao [2], Zheng Fang

Department of Physics, University of California San Diego

9500 Gilman Drive

La Jolla, CA 92093

and

Henry D.I. Abarbanel

Marine Physical Laboratory,

Scripps Institution of Oceanography

and

Department of Physics, University of California San Diego

9500 Gilman Drive

La Jolla, CA 92093

---

[1,2] These authors contributed equally to this research



## 1  Abstract

Statistical Data Assimilation (SDA) is the transfer of information from field or lab-
oratory observations to a user selected model of the dynamical system producing
those observations. The data is noisy and the model has errors; the informa-
tion transfer addresses properties of the conditional probability distribution of the
states of the model conditioned on the observations. The quantities of interest
in SDA are the conditional expected values of functions of the model state, and
these require the approximate evaluation of high dimensional integrals. We in-
troduce a conditional probability distribution and use the Laplace method with
annealing to identify the maxima of the conditional probability distribution. The
annealing method slowly increases the precision term of the model as it enters the
Laplace method. In this paper, we extend the idea of precision annealing (PA) to
Monte Carlo calculations of conditional expected values using Metropolis-Hastings
methods.

## 2  Introduction

We begin with a description of a framework within which we will discuss transfer
of information from data to a model of the processes producing the data.

Within an observation window in time, $[t_0 \leq t \leq t_F]$, we make a set of measure-
ments at times $t = \{\tau_1, \tau_2, ..., \tau_k, \tau_F\}$; $t_0 \leq \tau_k \leq t_F$. At each of these measurement
times, we observe $L$ quantities $\mathbf{y}(\tau_k) = \{y_1(\tau_k), y_2(\tau_k), ..., y_L(\tau_k)\}$. The number $L$
of observations at each measurement time $\tau_k$ is typically less, often much less, than
the number of degrees of freedom $D$ in the model of the observed system; $D \gg L$.

The quantitative characterization of the dynamical processes is through a
model we choose. It describes the interactions among the states of the observed
system. From the data $\{\mathbf{y}(\tau_k)\}$ we want to estimate the *unmeasured states* of the
model as a function of time as well as estimate any time independent physical
parameters in the model. At the end of the observation window $t = t_F$, we use the
estimated values of all model states and parameters to predict the model response
to new forcing of the system for $t \geq t_F$. The predictions are used to validate the
model (or not) as well as the estimation procedure.

The $D$-dimensional state of the model we call $x_a(t)$; $a = 1, 2, ..., D \geq L$. These
are selected by the user to describe the dynamical behavior of the observations
through a set of differential equations in continuous time

$$\frac{dx_a(t)}{dt} = F_a(\mathbf{x}(t), \mathbf{p}), .\tag{1}$$

Equivalently, in discrete time $t_n = t_0 + n\Delta t$; $n = 0, 1, ..., N$; $t_N = t_F$, the



dynamics is written as

$$x_a(t_{n+1}) = f_a(\mathbf{x}(t_n), \mathbf{p}) \ \text{ or } \ x_a(n+1) = f_a(\mathbf{x}(n), \mathbf{p}), \tag{2}$$

where $\mathbf{p}$ is a set of parameters, fixed in time, associated with the model. $\mathbf{f}(\mathbf{x}(n), \mathbf{p})$ is related to $\mathbf{F}(\mathbf{x}(t), \mathbf{p})$ through the choice the user makes for solving the continuous time flow for $\mathbf{x}(t)$ through a numerical solution method of choice (Press et al. 2007).

To make the discussion here a bit more compact, we will work henceforth in discrete time $t_n = t_0 + n\Delta t; \ n = 0, 1, ..., N; \ t_N = t_F$, and we will choose the observation times $\tau_k$ to be multiples of $\Delta t$ as well: $\tau_k = t_0 + k[n\tau]\Delta t; \ k = 1, 2, ..., F$.

As we proceed from the initiation of observations at $t_0$, we must use our model equations to move the state variables $\mathbf{x}(t_0) = \mathbf{x}(0)$, Eq. (2), from $t_0$ to $\tau_1 = t_0 + 1[n\tau]\Delta t$ where the first measurement is made. Then we use the model dynamics again to move along to $\tau_2 = t_0 + 2[n\tau]\Delta t$, where the second measurement is made, and so forth until we reach the time of the last measurement $t = \tau_F = t_0 + F[n\tau]\Delta t$ and finally move the model from $\mathbf{x}(\tau_F)$ to $\mathbf{x}(t_F)$.

We collect the $\mathbf{x}(t_n)$ for all $n$ into the **path** of the state of the model through $D$-dimensional space: $\mathbf{X} = \{\mathbf{x}(0), \mathbf{x}(1), ..., \mathbf{x}(n), ..., \mathbf{x}(N) = \mathbf{x}(F)\}$. The dimension of the path is $(N+1)D + N_p$, where $N_p$ is the number of parameters $\mathbf{p}$ in our model. In $\mathbf{X}$ we do not explicitly show the fixed parameters $\mathbf{p}$. This notation is illustrated in Fig. (1).

We now have two of the three required ingredients to effect our transfer of the information in the collection of all measurements $\mathbf{Y} = \{\mathbf{y}(\tau_1), \mathbf{y}(\tau_2), ..., \mathbf{y}(\tau_F)\}$ to the model $\mathbf{f}(\mathbf{x}(n), \mathbf{p})$ along the path $\mathbf{X}$ through the observation window $[t_0, t_F]$:

- (1) our noisy data $\mathbf{Y}$ and

- (2) a model of the processes producing the $\mathbf{Y}$. This model is devised by our experience and knowledge of those processes. The notation and a visual presentation of this is found in Fig. (1).

The **third** ingredient is comprised of methods to generate the transfer from $\mathbf{Y}$ to properties of the model. This will command our attention throughout this paper.

If the transfer methods are successful and, according to some metric of success, we arrange matters so that at the measurement times $\tau_k$, the $L$ model variables $\mathbf{x}(t)$ associated with $\mathbf{y}(\tau_k)$ are such that $x_l(\tau_k) \approx y_l(\tau_k); l = 1, 2, ..., L$, we are *not* finished. We have then only demonstrated that the model is consistent with the known data $\mathbf{Y}$. We must further use the model, completed by the estimates of the $\mathbf{p}$ and the state of the model at $t_F$, $\mathbf{x}(t_F)$, to predict forward for $t > t_F$, and we should succeed in comparison with measurements for $\mathbf{y}(\tau_r)$ for $\tau_r > t_F$. As the measure of success for predictions, we may use the same metric as utilized in the

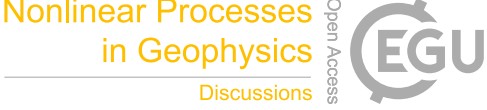



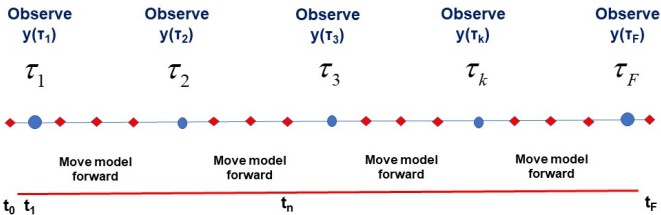

Figure 1: A visual representation of the time window $t_0 \leq t \leq t_F$ during which $L$-dimensional observations $\mathbf{y}(\tau_k)$ are performed at observation times $t = \tau_k$; $k = 1, , , ..., F$; $t_0 \leq \tau_k \leq t_F$. We also show times at which the $D$-dimensional model developed by the user $\mathbf{x}(n + 1) = \mathbf{f}(\mathbf{x}(n), \mathbf{p})$ is used to move forward from time $n$ to time $n + 1$: $t_n = t_0 + n\Delta t$; $n = 0, 1, ..., N$; $t_F = t_N$. $D \geq L$.




observation window. In the prediction window no further information from the
observations is passed to the model.

As a small aside, the same overall setup applies to supervised machine learning
networks (Abarbanel, Rozdeba, and Shirman 2018) where the observation window
is called the training set; the prediction window is called the test set, and prediction
is called generalization.

## 2.1   The Data are Noisy; the Model has Errors

Inevitably, the data we collect is noisy, and with equal assurance the model we
select to describe the production of those data has errors. This means we must,
at the outset, address a conditional probability distribution $P(\mathbf{X}|\mathbf{Y})$ as our goal
in the data assimilation transfer from $\mathbf{Y}$ to the model. In Abarbanel 2013, we
describe how to use the Markov nature of the model dynamics $\mathbf{x}(n) \rightarrow \mathbf{x}(n+1) =$
$\mathbf{f}(\mathbf{x}(n), \mathbf{p})$ and the definition of conditional probabilities to derive the recursion
relation connecting observations and dynamics at times $t_{n+1}$ and $t_n$:

$$
\begin{aligned}
P(\mathbf{X}(n+1)|\mathbf{Y}(n+1)) &= \frac{P(\mathbf{y}(n+1), \mathbf{x}(n+1), \mathbf{X}(n)|\mathbf{Y}(n))}{P(\mathbf{y}(n+1)|\mathbf{Y}(n))\, P(\mathbf{x}(n+1), \mathbf{X}(n+1)|\mathbf{Y}(n))} \bullet \\
&\quad P(\mathbf{x}(n+1)|\mathbf{x}(n)) P(\mathbf{X}(n)|\mathbf{Y}(n)) \\
&= \exp[CMI(\mathbf{y}(n+1), \mathbf{x}(n+1), \mathbf{X}(n)|\mathbf{Y}(n))] \bullet \\
&= \frac{P(\mathbf{y}(n+1)|\mathbf{x}(n+1), \mathbf{X}(n), \mathbf{Y}(n))}{P(\mathbf{y}(n+1)|\mathbf{Y}(n))} \bullet \\
&\quad P(\mathbf{x}(n+1)|\mathbf{x}(n)) P(\mathbf{X}(n)|\mathbf{Y}(n)),
\end{aligned}
\tag{3}
$$

where we have identified $CMI(a, b|c) = \log[\frac{P(a,b|c)}{P(a|c)\,P(a|c)}]$. This is Shannon's con-
ditional mutual information (Fano 1961) telling us how many bits (for $\log_2$) we
know about $a$ when observing $b$ conditioned on $c$. For us $a = \{\mathbf{y}(n+1)\}, b =$
$\{\mathbf{x}(n+1), \mathbf{X}(n+1)\}, c = \{\mathbf{Y}(n)\}$.

Using this recursion relation to move backwards from the end of the observation
window from $t_F = t_0 + N\Delta t$ through the measurements at times $\tau_k$ to the start of
the window at $t_0$, we may write, up to factors independent of $\mathbf{X}$

$$
P(\mathbf{X}|\mathbf{Y}) = \left\{ \prod_{k=1}^{F} P(\mathbf{y}(\tau_k)|\mathbf{X}(\tau_k), \mathbf{Y}(k-1)) \prod_{n=0}^{F-1} P(\mathbf{x}(n+1)|\mathbf{x}(n)) \right\} P(\mathbf{x}(0)).
\tag{4}
$$

If we now write $P(\mathbf{X}|\mathbf{Y}) \propto \exp[-A(\mathbf{X})]$. $A(\mathbf{X})$, the negative of the log likelihood,
we call the action. Conditional expected values for functions $G(\mathbf{X})$ along the path
$\mathbf{X}$ are defined by

$$
E[G(\mathbf{X})|\mathbf{Y}] = \langle G(\mathbf{X}) \rangle = \frac{\int d\mathbf{X}\, G(\mathbf{X}) e^{-A(\mathbf{X})}}{\int d\mathbf{X}\, e^{-A(\mathbf{X})}},
\tag{5}
$$




$dX = \prod_{n=0}^{N} d^D x(n)$, and all factors in the action independent of $\mathbf{X}$ cancel out here. The action takes the convenient expression

$$A(\mathbf{X}) = -\sum_{k=1}^{F}\left\{ \log[P(\mathbf{y}(\tau_k)|\mathbf{X}(\tau_k),\mathbf{Y}(k-1))] - \sum_{n=0}^{N} \log[P(\mathbf{x}(n+1)|\mathbf{x}(n))]\right\} - \log[P(\mathbf{x}(0))], \tag{6}$$

which is the sum of the terms which modify the conditional probability distribution when an observation is made at $t = \tau_k$ and the sum of the stochastic version of $\mathbf{x}(n) \to \mathbf{x}(n+1) - \mathbf{f}(\mathbf{x}(n),\mathbf{p})$ and finally the distribution when the observation window opens at $t_0$.

What quantities $G(\mathbf{X})$ are of interest? One natural one is the path of model states and parameters $G(\mathbf{X}) = \mathbf{X}_\mu; \mu = \{a,n\}; a = 1,2,...,D; n = 0,1,2,...N$ itself; another is the covariance around that mean $\langle \mathbf{X}_\mu \rangle = \bar{\mathbf{X}}_\mu : \langle (\mathbf{X}_\mu - \bar{\mathbf{X}}_\mu)(\mathbf{X}_\nu - \bar{\mathbf{X}}_\nu) \rangle$. Other moments are of interest, of course. If one has an anticipated form for the distribution at large $\mathbf{X}$, then $G(\mathbf{X})$ may be chosen as a parametrized version of that form and those parameters determined near the maximum of $P(\mathbf{X}|\mathbf{Y})$.

The action simplifies to what we call the 'standard model' of data assimilation when (1) observations $\mathbf{y}$ are related to their model counterparts via Gaussian noise with zero mean and diagonal precision matrix $\mathbf{R}_m$, and (2) model errors are associated with Gaussian errors of mean zero and diagonal precision matrix $\mathbf{R}_f$:

$$A(\mathbf{X}) = \sum_{k=1}^{F}\sum_{l=1}^{L}\frac{R_m}{2}(x_l(\tau_k)-y_l(\tau_k))^2 + \sum_{n=0}^{N-1}\sum_{a=1}^{D}\frac{R_f(a)}{2}(x_a(n+1)-f_a(\mathbf{x}(n),\mathbf{p}))^2. \tag{7}$$

If we have knowledge of the distribution $P(\mathbf{x}(0))$ at $t_0$ we may add it to this action, Eq. (6). If we have no knowledge of $P(\mathbf{x}(0))$, we may take its distribution to be uniform over the dynamic range of the model variables, then it, as here, is absent, canceling numerator and denominator in Eq. (5).

## 2.2 The Goal of SDA

Our challenge is to perform integrals such as Eq. (5). One should anticipate that the dominant contribution to the expected value comes from the maxima of $P(\mathbf{X}|\mathbf{Y})$ or, equivalently the minima of $A(\mathbf{X})$.

We note, as before, that when $\mathbf{f}(\mathbf{x}(n),\mathbf{p})$ is nonlinear in $\mathbf{X}$, as it always is in interesting examples, the expected value integral Eq. (5) is not Gaussian. So, some thinking is in order before approximating this high dimensional integral. We turn to that now. After consideration of methods to do the integral, we will return to an example taken from an instructional model often used in the geosciences.



Two generally useful methods available for evaluating this kind of high-dimensional integral are Laplace's method (Laplace 1774; Laplace 1986) and Monte Carlo techniques (Press et al. 2007; Kostuk et al. 2012; Neal 2011). The Laplace methods, including the idea of precision annealing for the model error term are discussed in Quinn 2010; Ye 2016; Ye, Rey, et al. 2015; Ye, Kadakia, et al. 2015.

The drawbacks of using Laplace methods, including precision annealing methods, include the need for evaluating very high dimensional derivatives of $A(\mathbf{X})$ with respect to $\mathbf{X}$ and using them in the nonlinear optimization algorithms selected. Further, when successful in identifying the path yielding the smallest value of $A(\mathbf{X})$, thus the potentially dominant contribution to Eq. (5), we do not sample the desired conditional probability distribution away from its maximum. Evaluating corrections to the leading Laplace contributions is familiar as perturbation theory in statistical physics. The convergence of such perturbation methods can depend sensitively on the functional form of the action in $\mathbf{X}$.

We now turn to extending the annealing techniques that explore the variation of $\langle G(\mathbf{X}) \rangle$ in the magnitude of the precision matrix $\mathbf{R}_f$ for the model error from Laplace's method to Monte Carlo methods for approximating the path integral for $\langle G(\mathbf{X}) \rangle$.

# 3 Precision Annealing Monte Carlo Methods

Monte Carlo methods for the approximate evaluation of quantities such as $\langle G(\mathbf{X}) \rangle$ via Eq. (5) have been intensively explored and utilized for decades (Metropolis et al. 1953; Hastings 1970; Neal 2011).

Standard MC calculations, following many years of developments from Metropolis et al. 1953; Hastings 1970, seek to estimate the conditional probability distribution $P(\mathbf{X}|\mathbf{Y})$ by starting somewhere in **path** space $\mathbf{X}[\text{init}]$, making moves in path space from this initial path and accepting and rejecting proposed moves according to a criterion based on detailed balance.

The folklore about these calculations is that one can begin more-or-less anywhere in path space and after a large enough number of steps leading to rejected paths and accepted paths proceeding from $\mathbf{X}[\text{init}]$, one will arrive at a good expected value in Eq. (5). Indeed the error is order the inverse square root of the number of accepted paths with the numerator essentially the variance in the function $G(\mathbf{X})$ whose expected value one wishes to estimate.

In practice, if one can choose $\mathbf{X}[\text{init}]$ 'close' to the maximum of $P(\mathbf{X}|\mathbf{Y})$ the more efficient the procedure is expected to be; namely high accuracy may be achieved with fewer steps. Of course, if we knew where the maximum of $P(\mathbf{X}|\mathbf{Y})$ were located (Shirman 2018), we'd start there and sample, through proposals for acceptable paths, a sufficient neighborhood of that minimum action path to arrive



at a good estimation of $\langle G(\mathbf{X}) \rangle$. It is not hard to see that as we do **not** know the global minimum of the action, there is a lot of room for algorithms that make good proposals for new acceptable paths and clever choices for $\mathbf{X}$[init].

Our idea in this paper is to follow the suggestions of Quinn 2010; Ye 2016; Ye, Rey, et al. 2015; Ye, Kadakia, et al. 2015 about how we can 'anneal' the precision of the model error term of the action starting with $R_f = 0$, at which the global minimum of the standard model action is clear. From there, we slowly increase $R_f$ until it is very large and imposes the underlying dynamical model more and more precisely. This method was developed in the context of Laplace approximations to the expected value integrals (Quinn 2010; Ye 2016; Ye, Rey, et al. 2015; Ye, Kadakia, et al. 2015) and has been extensively tested in several areas of application of SDA.

## 3.1 $R_f = 0$; Choosing Initial Paths $\mathbf{X}^q$[init]; $q = 1, 2, ..., N_I$ for the PAMC Procedure

Our strategy in this paper is to vary the 'hyperparameter' $R_f$ that sets the scale for the precision of the model error term in Eq. (7). When $R_f \to \infty$ the model is very precise and deterministic.

In our precision annealing strategy, we start at the other end of the scale where $R_f = 0$. At this value the model error term is absent, and the 'standard' model action is quadratic in the measured variables $x_l(n)$. At $R_f = 0$ the action is a minimum when we select $x_l(\tau_k = t_0 + k[n\tau]\Delta t) = y_l(\tau_k)$; $l = 1, 2, ..., L$. This is the global minimum of the action at $R_f = 0$, and it is quite degenerate as the action does not depend on the unmeasured model state variables or the parameters in the model.

The path of the model state (not showing the $N_p$ fixed parameters $\mathbf{p}$) is comprised of

$$\mathbf{X} = \{x_1(0), x_2(0), ..., x_D(0), x_1(1), x_2(1), ..., x_D(1), \ldots x_1(N), x_2(N), ..., x_D(N)\}. \tag{8}$$

In our $N_I$ initial paths for the Monte Carlo search, $\mathbf{X}^q$[init], we always choose $x_l(\tau_k = t_0 + [n\tau k]\Delta t) = y_l(\tau_k)$; $l = 1, 2, ..., L$, and we wish to select the other components of $\mathbf{X}$[init] in a manner that is 'close' to a minimum action path. We select $q = 1, 2, ..., N_I$ initial paths $\mathbf{X}^q$[init] so we will be tracking an *ensemble* of paths using various Monte Carlo protocols.

To complete our choice of initial paths, we now split the state variables $x_a(n)$ into those observed $a = 1, 2, ..., L$ and those unobserved $a > L$. The latter we call the 'rest' and write them as $x_R(n)$; $R = L + 1, L + 2, ..., D$. The dynamical equations (in discrete time) can now be written

$$x_l(n + 1) = f_l(x_l(n), x_R(n)) \quad x_R(n + 1) = f_R(x_l(n), x_R(n)). \tag{9}$$





Starting with any initial condition $\{x_l^q(0), x_R^q(0)\}$ we generate solutions to these
dynamical equations by using Eq. (9) . We proceed by choosing $q = 1, 2, ..., N_I$
initial conditions $\{x_l^q(0), x_R^q(0)\}$ from a uniform distribution over the ranges of
$\{x_l(0), x_R(0)\}$ which we can infer from the data and from forward integration
of the model. Using the $N_I$ $\{x_l^q(0), x_R^q(0)\}$ we generate $N_I$ paths. However, we
substitute for $x_l(t_0 + k[n\tau])$, **whenever** it occurs in the equations Eq. (9), the
observed value $y_l(\tau_k = t_0 + k[n\tau]\Delta t) = x_l(t_0 + k[n\tau])$.

This generates $q = 1, 2, ..., N_I$ initial paths $\mathbf{X}^q[\text{init}]$, one from each selection
of $\{x_l^q(0), x_R^q(0)\}$, everyone of which has zero standard action. Each of these
paths corresponds to an initial action at the global minimum for $R_f = 0$, namely
$A(X^q[\text{init}]) = 0$.

## 3.2  Precision Annealing Procedure

We next move from $R_f = 0 \rightarrow R_{f0} > 0$ and using the $N_I$ $\mathbf{X}^q[\text{init}]$ paths, perform
an MCMC procedure.

Our first procedure is to use a fixed number of iterations of Metropolis-Hastings
(M-H) proposals/acceptance steps comprised of a fixed number of "burn-in" steps
followed by a fixed number of iteration steps. The M-H step size is changed as we
go along to assure a good acceptance rate.

At the termination of the M-H steps, we will have $j = 1, 2, ..., N_A(q, 0)$ accepted
paths $\mathbf{X}_j^q[\text{init}]$ for each of the $q = 1, 2, ..., N_I$ initial paths. We use these $N_A(q, 0)$
accepted paths to estimate $N_I$ expected paths $\bar{\mathbf{X}}^q[0]$ using

$$\bar{\mathbf{X}}^q[0] = \frac{1}{N_A(q, 0)} \sum_{j=1}^{N_A(q,0)} \mathbf{X}_j^q[\text{init}]. \tag{10}$$

These $N_I$ paths, $\bar{\mathbf{X}}^q[0]$, evaluated at $R_f = R_{f0}\alpha^0$ are set aside and retained for use
as initial paths for the next step in the PA procedure. This completes the first
step of the PAMC process; $R_f = R_{f0}\alpha^0$ at this step.

The PA strategy is exposed now: at $R_f = 0$ choose a dynamically selected set
of $N_I$ initial paths $\mathbf{X}^q[\text{init}]$. All these paths have zero action. Then raise the value
of $R_f$ to a small positive number $R_f \rightarrow R_{f0} > 0$, thus introducing the model error
into the action, but keeping $R_f$ quite small, and at this value of $R_f$ use the $N_I$
paths $\mathbf{X}^q[\text{init}]$ in the selected M-H procedure resulting in a set of paths 'near' the
$\mathbf{X}^q[\text{init}]$ as $R_f$ is small. The resulting $N_I$ paths at this small value of $R_f$ are then
used as initial paths when we raise $R_f \rightarrow R_{f0}\alpha$. This sequential use of accepted
paths from the previous value of $R_f$ comprises the precision annealing approach.
Now we describe this in a bit more detail.

As the second step in PAMC we move $R_f$ from $R_{f0} \rightarrow R_{f0}\alpha^1$ with $\alpha > 1$. At
this increased value of $R_f$ we use the same MCMC procedure but now starting at



the $\bar{\mathbf{X}}^q[0]$ as $N_I$ initial paths. This results in $j = 1, 2, ..., N_A(q, 1)$ accepted paths $\bar{\mathbf{X}}_j^q[0]$ for each $q$. Again we form $N_I$ expected paths using

$$\bar{\mathbf{X}}^q[1] = \frac{1}{N_A(q, 1)} \sum_{j=1}^{N_A(q,1)} \bar{\mathbf{X}}_j^q[0]. \tag{11}$$

This completes the second step of the PAMC process; $R_f = R_{f0}\alpha^1$ at this step.

Next we move $R_f$ from $R_{f0}\alpha^1 \to R_{f0}\alpha^2$ with $\alpha > 1$. At this increased value of $R_f$ we use the same MCMC procedure but now starting at the $\bar{\mathbf{X}}^q[1]$ as $N_I$ initial paths. This results in $j = 1, 2, ..., N_A(q, 2)$ accepted paths $\bar{\mathbf{X}}_j^q[1]$ for each $q$. Again we form $N_I$ expected paths

$$\bar{\mathbf{X}}^q[2] = \frac{1}{N_A(q, 2)} \sum_{j=1}^{N_A(q,2)} \bar{\mathbf{X}}_j^q[1]. \tag{12}$$

This completes the third step of the PAMC process; $R_f = R_{f0}\alpha^2$ at this step.

Continue on in this manner increasing the value of $R_f$ from $R_f = R_{f0}\alpha^{\beta-1}$ to $R_f = R_{f0}\alpha^\beta$. At this new value of $R_f$ we use the same MCMC procedure but now starting at the $\bar{\mathbf{X}}^q[\beta - 1]$ as $N_I$ initial paths. This results in $j = 1, 2, ..., N_A(q, \beta)$ accepted paths $\bar{\mathbf{X}}_j^q[\beta]$ for each $q$. Form the $N_I$ expected paths

$$\bar{\mathbf{X}}^q[\beta] = \frac{1}{N_A(q, \beta)} \sum_{j=1}^{N_A(q,\beta)} \bar{\mathbf{X}}_j^q[\beta - 1]. \tag{13}$$

This completes the $\beta^{th}$ step of the PAMC process; $R_f = R_{f0}\alpha^\beta$ at this step.

This 'stepping in $\beta$' continues until $\beta$ is 'large enough'; we will discuss a criterion for that shortly. At this value of 'large enough' $\beta$, we will have performed the MCMC procedure one last time (at $R_f = R_{f0}\alpha^\beta$) to collect, for each $q$, $N_A(q, \beta)$ accepted paths $\bar{\mathbf{X}}[\beta]_j$; $j = 1, 2, ..., N_A(q, \beta)$.

Finally, we estimate $\langle G(\mathbf{X}) \rangle$ as the average (expected value) over the $N_I$ paths reached at $R_f = R_{f0}\alpha^\beta$

$$\langle G(\mathbf{X}) \rangle = \frac{1}{N_I} \sum_{q=1}^{N_I} G(\bar{\mathbf{X}}^q[\beta]), \tag{14}$$

and this completes our PA Monte Carlo procedure. Note that at each increment of $\beta$ we use as initial paths the $N_I$ expected paths from the previous $\beta$.

We evaluate the action on each of the $N_I$ paths at each value of $R_f$ and plot $A(\mathbf{X}^q)$ versus $\log[R_f/R_{f0}]$. In such an 'action level' plot, as the precision of the


<sup>271</sup> model is increased, if the model is consistent with the data and the number of
<sup>272</sup> observed measurements, $L$, at each $\tau_k$ is large enough, the action level plot values
<sup>273</sup> will become independent of $R_f$ and one will stand out as lower than the rest. The
<sup>274</sup> path corresponding to that lowest action level will dominate the expected value
<sup>275</sup> integral of interest.

<sup>276</sup> We will see this happen in the example discussed in the next section. It also
<sup>277</sup> happens in the Laplace approximation to finding the largest values of $P(\mathbf{X}|\mathbf{Y})$
<sup>278</sup> (Quinn 2010; Ye 2016; Ye, Rey, et al. 2015; Ye, Kadakia, et al. 2015). The interpre-
<sup>279</sup> tation of this transition is that the number of directions in model state space that
<sup>280</sup> are explored by the $L$, independent measurements at each $\tau_k$, $y_l(\tau_k)$; $l = 1, 2, ..., L$
<sup>281</sup> reveal, and through the estimation procedure (PAMC), 'cure' the intrinsic local
<sup>282</sup> unstable directions in the nonlinear model $\mathbf{x}(n + 1) = \mathbf{f}(\mathbf{x}(n), \mathbf{p})$. This happens
<sup>283</sup> with higher precision as $R_f$ becomes larger and larger.

## <sup>284</sup> 4    Example of PAMC Calculations

<sup>285</sup> We explore the instructional model from Lorenz 2006, widely used in numerical
<sup>286</sup> weather prediction analyses, as a test bed for methods of data assimilation. This
<sup>287</sup> model has a $D$-dimensional state variable $\mathbf{x}(t) = \{x_1(t), x_2(t), ..., x_D(t)\}$ satisfying

$$\frac{dx_a(t)}{dt} = x_{a-1}(t)[x_{a+1} - x_{a-2}(t)] - x_a(t) + \nu \ \ a = 1, 2, ..., D, \qquad (15)$$

<sup>288</sup> in which $x_{-1}(t) = x_{D-1}(t)$, $x_0(t) = x_D(t)$, and $x_1(t) = x_{D+1}(t)$. $\nu$ is a constant
<sup>289</sup> forcing term; the solutions of these equations for $D \geq 4$ are chaotic when $\nu \approx 8.0$ or
<sup>290</sup> more. We will report on calculations with $D = 5$ and with $D = 20$ with $\nu = 8.17$.

<sup>291</sup> Our numerical calculations are 'twin experiments' in which for a selected $D$ we
<sup>292</sup> choose $\mathbf{x}(t_0) = \mathbf{x}(0)$ and using a time step $\Delta t = 0.025$ generate solutions $\mathbf{x}(t)$ over
<sup>293</sup> an observation window $[t_0, t_F] : t_0 \leq t \leq t_0 + N\Delta t = t_F$. To each $x_a(t)$ we add
<sup>294</sup> Gaussian noise with mean zero and variance $\sigma^2$, these now comprise our library
<sup>295</sup> of 'observed data;' $y_a(t) = x_a(t) + \sigma N(0, 1)$. We then select $L \leq D$ of these noisy
<sup>296</sup> data, and form the action

$$A(\mathbf{X}) = \sum_{n=0}^{N} \sum_{l=1}^{L} \frac{R_m(n)}{2} (y_l(n) - x_l(n))^2 + \frac{R_f}{2} \sum_{n=0}^{N-1} \sum_{a=1}^{D} [x_a(n+1) - f_a(\mathbf{x}(n))]^2, \ (16)$$

<sup>297</sup> and $R_m(n)$ is nonzero only when there is a measurement at $t_n$, and at each of these
<sup>298</sup> times $L$ quantities are observed. The first term on the right in Eq. (16) is the
<sup>299</sup> measurement error, and the second, the model error.

<sup>300</sup> Our calculations were performed with the choices: $D = 20$, $\alpha = 1.4$, $R_{f0} = 1.0$,
<sup>301</sup> $R_m = 1.0$, $N_I = 50$, $\Delta t = 0.025$, and various choices of $L$ from 5 to 12.



In Fig. (2) we display the action levels as a function of $\beta$ at $L = 5$. We can see that PAMC identifies many action levels, corresponding to many peaks in the conditional probability distribution $P(\mathbf{X}|\mathbf{Y}) \propto \exp[-A(\mathbf{X})]$, Eq. (16). From $\beta \approx 30$ we see one level moving away from the collection of larger action levels as $\beta$ increases. However, no action level has become essentially independent of $R_f$ suggesting that the accuracy with which the model is enforced remains too small. We expect that as the number of measurements at each $\tau_k$ is increased more information about the phase space instabilities will be passed from the data to the model and that the structure of the action level plot will change.

In Fig. (3) we now display the action levels and its components, the measurement errors and the model errors, when $L = 12$. Here the behavior of the action levels is quite different. The model error decreases over a large range of $R_f$ until the numerical stability of the evaluation of this term is reduced as small errors in $\mathbf{x}(n + 1) - \mathbf{f}(\mathbf{x}(n), \mathbf{p})$ are magnified by large values of $R_f$. As this result appears, the action for each of the $N_I$ paths at each $\beta$ levels off, becoming essentially independent of $R_f$, and matches the measurement error, as it must do for consistency (Quinn 2010; Ye 2016; Ye, Rey, et al. 2015; Ye, Kadakia, et al. 2015).

The PAMC procedure, as does the Laplace approximation version of precision annealing (Quinn 2010; Ye 2016; Ye, Rey, et al. 2015; Ye, Kadakia, et al. 2015), permits the estimation of the parameter $\nu$ at each value of $\beta$. In Fig. (4) we display all $N_I = 50$ estimated values of $\nu$ at each value of $\beta$. As PAMC is an ensemble method sampling in the neighborhood of a peak (or peaks) of the conditional probability distribution, we do not arrive at a single value for $\nu$. Taking the $N_I$ values of $\nu(\beta)$ and evaluating the means and standard deviation at each $\beta$, we show the result in Fig. (5) in which it is clear that the estimated value of $\nu$ becomes essentially independent of $\beta$ for $\beta \approx 40$ and larger.

Until this point we have examined outcomes of the PAMC estimation procedure. All of the state variables, **measured** and **unmeasured**, as well as the forcing parameter were reported over the observation window $[0 \leq t \leq 5.0]$. In a 'twin experiment' as here, we have generated the data by solving a known dynamical equation and adding noise to the output of the $D = 20$ times series with a known value of $\nu$. The point of a twin experiment is to test the method of transfer of information in SDA. As we have $D - L$ **unobserved** state variables at each $L$, and an **unobserved** parameter $\nu$, the only tool to determine how well the estimation procedure has done in its task is to predict for $t > 5$ into a prediction window where no information from observations is passed back from the model. We now examine how well the estimation has been performed by predicting both an observed and an unobserved time series among the $D$ available. We already see from Fig. (5) that the input value of $\nu = 8.17$ has accurately been estimated; the apparent bias in this parameter estimation has also been seen in earlier Monte Carlo


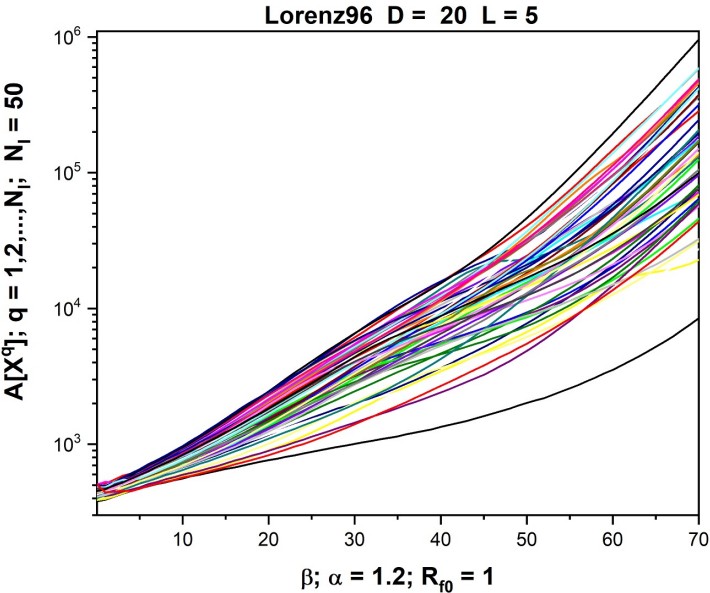

Figure 2: The values of the actions Eq. (16) for the $D = 20$ dimensional Lorenz96 model when $L = 5$ of the dynamical variables $\mathbf{x}(t)$ are observed. The actions are evaluated as a function of $\beta = \log_\alpha[R_f/R_f0]$ where $\alpha = 1.4$ and $R_{f0} = 1.0$. We perform the Precision Annealing Monte Carlo (PAMC) calculation starting with $N_I$ initial paths at each $R_f$. We used $N_I = 50$ in these calculations. Displayed here are $N_I$ action values at each $R_f$ (or $\beta$). These actions are evaluated along the expected path resulting from the accepted paths generated during the Metropolis-Hastings procedures from each of the $N_I$ initial paths.



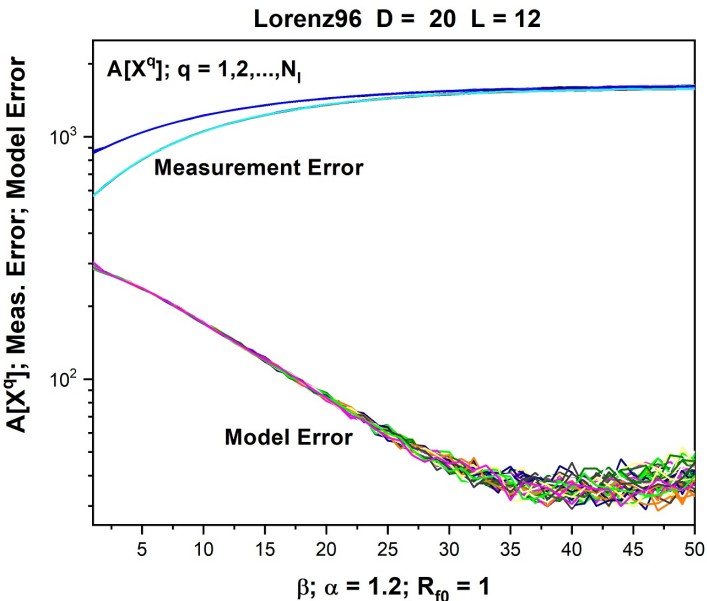

Figure 3: The values of the actions Eq. (16), the measurement error, and the model error for the $D = 20$ dimensional Lorenz96 model when $L = 12$ of the dynamical variables $\mathbf{x}(t)$ are observed; the observed variables are $[x_1(t), x_2(t), x_4(t), x_6(t), x_7(t), x_9(t), x_{11}(t), x_{12}(t), x_{14}(t), x_{16}(t), x_{17}(t), x_{19}(t)]$. The actions, the measurement error, and the model error are evaluated as a function of $\beta = \log_\alpha[R_f/R_f 0]$ where $\alpha = 1.4$ and $R_{f0} = 1.0$. We perform the Precision Annealing Monte Carlo (PAMC) calculation starting with $N_I$ initial paths at each $R_f$. We used $N_I = 50$ in these calculations; on display here are $N_I$ action, measurement error, and model error values at each $R_f$ (or $\beta$). These are evaluated along the expected path resulting from the accepted paths generated during the Metropolis-Hastings procedures from each of the $N_I$ initial paths. In this case, when $L = 12$, the model error becomes much smaller than the measurement error as $\beta$ is increased. This leads the action to become effectively equal to the action itself and essentially independent of $R_f$. We have seen this before in the precision annealing variational principle calculations (Quinn 2010; Ye 2016; Ye, Rey, et al. 2015; Ye, Kadakia, et al. 2015).





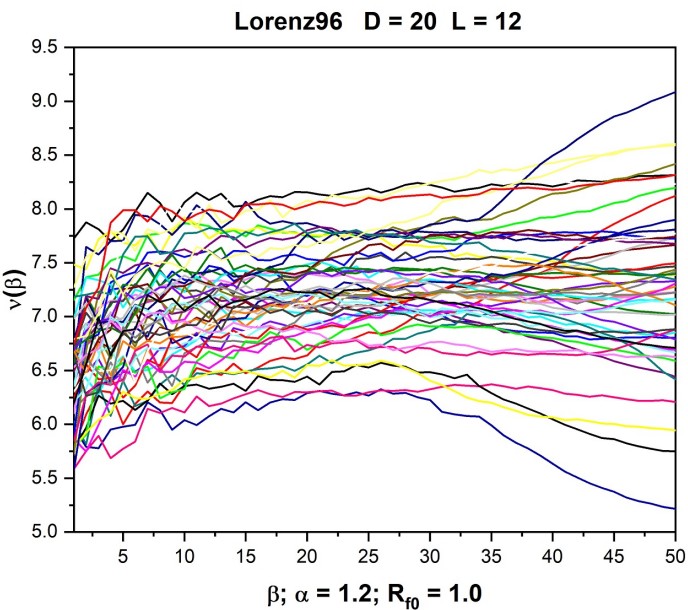

Figure 4: The values of the Lorenz96 model forcing parameter $\nu$ at each value of $\beta$ for each of the $N_I$ paths associated with the $N_I$ Metropolis-Hastings procedures from each of the $N_I$ initial paths.

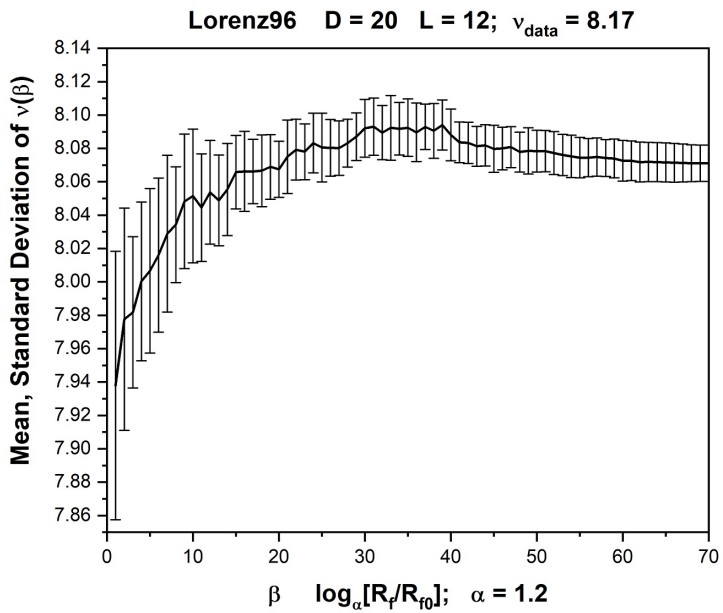

Figure 5: The estimated parameter in the Lorenz96, D = 20 data when L = 12.
The mean and standard deviation of $\nu$ at each $\beta$ is shown.



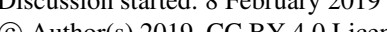


twin experiment Kostuk et al. 2012; Kostuk 2012, and its origins are discussed there.

Fig. (6) shows the **observed** model variable $x_2(t)$ for the Lorenz96 model with $D = 20, L = 12$ and $\Delta t = 0.025$. The noisy data from solutions of the model equations from the 'observed' variables $[1, 2, 4, 6, 7, 9, 11, 12, 14, 16, 17, 19]$. The estimation of $x_2(t)$ during the observation window using PAMC to transfer information from the data to the model is shown in red, and the prediction using all the estimated states of the model, $\mathbf{x}(t = 5)$, and the estimated model parameter, is shown in green $\mathbf{x}(t \geq 5)$. Our knowledge of this dynamical system (Kostuk 2012) indicates that the largest global Lyapunov exponent is approximately 1.2 in the time units indicated by $\Delta t$. The deviation of the predicted trajectory $x_2(t)$ from $t \approx 6.0$ is consistent with the accuracy of the estimated state $\mathbf{x}(t)$ and this Lyapunov exponent.

Fig. (7) shows the **unobserved** model variable $x_{20}(t)$ for the Lorenz96 model with $D = 20, L = 12$ and $\Delta t = 0.025$. The noisy data from solutions of the model equations from the 'observed' variables $[1, 2, 4, 6, 7, 9, 11, 12, 14, 16, 17, 19]$. The estimation of $x_{20}(t)$ during the observation window using PAMC to transfer information from the data to the model is shown in red, and the prediction using all the estimated states of the model, $\mathbf{x}(t = 5)$, and the estimated model parameter, is shown in blue $\mathbf{x}(t \geq 5)$. Our knowledge of this dynamical system (Kostuk 2012) indicates that the largest global Lyapunov exponent is approximately 1.2 in the time units indicated by $\Delta t$. The deviation of the predicted trajectory $x_{20}(t)$ from $t \approx 6.4$ is consistent with the accuracy of the estimated state $\mathbf{x}(t)$ and this Lyapunov exponent.

# 5 Discussion and Summary

In statistical data assimilation, one transfers information from a set of noisy data $\mathbf{Y}$ to models of the observations. The models have errors and the probability $P(\mathbf{X}|\mathbf{Y})$ of the model states, conditioned on the data, plays a central role. From this conditional probability distribution, we want to approximate conditional expected values of functions $G(\mathbf{X})$ on the model state

$$E[G(\mathbf{X})|\mathbf{Y}] = \int d\mathbf{X} P(\mathbf{X}|\mathbf{Y})G(\mathbf{X}) = \frac{\int d\mathbf{X} \exp[-A(\mathbf{X})]G(\mathbf{X})}{\int d\mathbf{X} \exp[-A(\mathbf{X})]}, \qquad (17)$$

where $A(\mathbf{X}) \propto -\log[P(\mathbf{X}|\mathbf{Y})]$ is the 'action' associated with the information transfer process during an observation window in time, when the information transfer occurs. Observations of the dynamical system underlying the measurements may be sparse; the number of measurements one is able to accomplish at any moment in time is typically small compared to the degrees of freedom in the model.


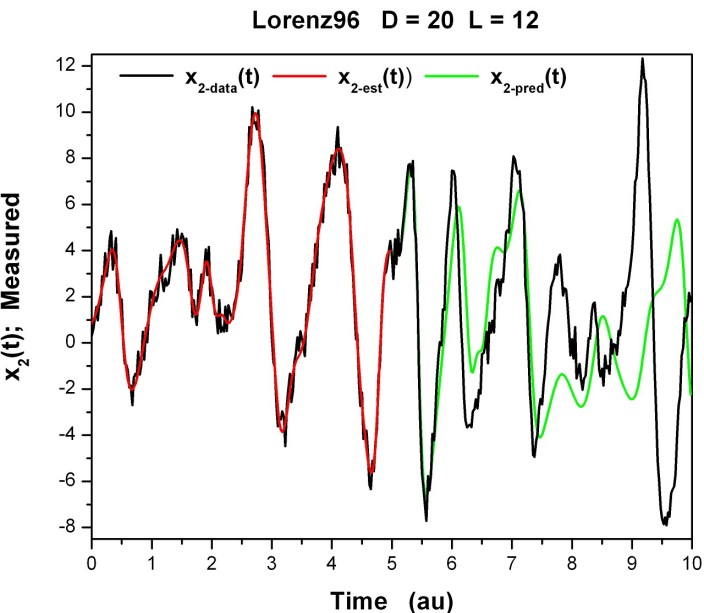

Figure 6: We display the **observed** dynamical variable $x_2(t)$ for the time interval $0 \leq t \leq 10.0$. In black is the full set of data. In red is the estimated $x_2(t)$ over the observation window $0 \leq t \leq 5.0$, and in green is the predicted $x_2(t)$ over the prediction window $5.0 < t \leq 10.0$. The prediction uses the values of $\mathbf{x}(t = 5.0)$ for the full estimated state at the end of the observation window as well as the parameter $\nu$ estimated in the PAMC procedure. This calculation uses the Lorenz96 model with D = 20 and L = 12. $\Delta t = 0.025$.

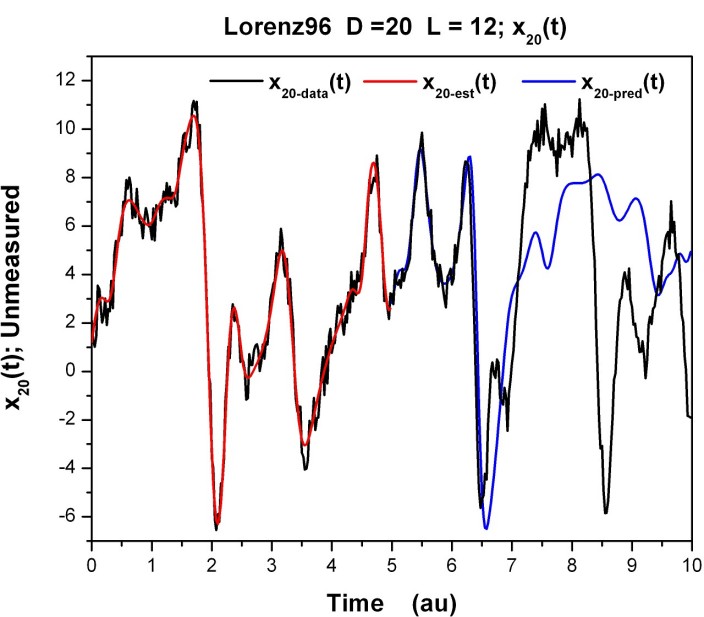

Figure 7: We display the **unobserved** dynamical variable $x_{20}(t)$ for the time interval $0 \leq t \leq 10.0$. In black is the full set of data. In red is the estimated $x_{20}(t)$ over the observation window $0 \leq t \leq 5.0$, and in blue is the predicted $x_{20}(t)$ over the prediction window $5.0 < t \leq 10.0$. The prediction uses the values of $\mathbf{x}(t = 5.0)$ for the full estimated state at the end of the observation window as well as the parameter $\nu$ estimated in the PAMC procedure. This calculation uses the Lorenz96 model with D = 20 and L = 12. $\Delta t = 0.025$.


However, one requires some approximate knowledge of the full state of the model at the final time-point of the observation window. This means one must estimate the **unmeasured** model state variables as well as any unknown time independent model parameters, then validate the model with predictions for times after the observation window.

In this paper we have addressed approximating such integrals using a precision annealing Monte Carlo method. In the context of a model $\mathbf{x}(t_{n+1}) = \mathbf{f}(\mathbf{x}(t_n), \mathbf{p})$ and observations $y_l(\tau_k)$ at times $t_0 \leq \tau_k \leq t_F$ (with $t_F = t_0 + N\Delta t$), the action reflects Gaussian errors of the measurements and of the nonlinear model, given by

$$A(\mathbf{X}) = \sum_{n=0}^{N} \sum_{l=1}^{L} \frac{R_m(n)}{2} (y_l(n) - x_l(n))^2 + \frac{R_f}{2} \sum_{n=0}^{N-1} \sum_{a=1}^{D} [x_a(n+1) - f_a(\mathbf{x}(n))]^2, \quad (18)$$

where $R_m(n)$ is nonzero only when there is a measurement at $t_n$. The precision of the model error is $R_f$ and the annealing procedure is initiated at $R_f$ very small, then continued to a very large $R_f$. The core idea is that when $R_f$ is small, the global minimum of $A(\mathbf{X})$ is easily identifiable where $x_l(\tau_k) \approx y_l(\tau_k)$. Increasing $R_f$ slowly allows one to track the global minimum as the nonlinearity in the action plays a more and more significant role.

The details of this PAMC procedure, implemented by a Metropolis-Hastings Monte Carlo method at each $R_f$, are given as a general outline. We then present results in detail for an instructional model - the Lorenz 1996 equations (Lorenz 2006), widely used to explore geophysical SDA methods.

In addition to the PAMC method, we introduce an initialization method for selecting a starting point in **path** space $\mathbf{X}$. From this starting point, we begin to make proposals and accept new samples in order to evaluate the conditional probability distribution.

Our PAMC methods are clearly not restricted to the specific example we used to demonstrate its operation, nor is the use of a Metropolis-Hastings procedure required in its implementation. We will follow this paper with one describing the use of a Hamiltonian Monte Carlo (HMC) procedure (Duane et al. 1987; Neal 2011; Betancourt 2018).

How is one to choose between the use of a precision annealing method for the Laplace approximation to expected value integrals and Monte Carlo methods (Metropolis-Hastings or HMC)? The key difference among the methods is that the Metropolis-Hastings Monte-Carlo does not require carrying along Jacobians or Hessians of the action $A(\mathbf{X})$ and samples the conditional probability distribution with paths $\mathbf{X}$ in model state space. The Laplace method requires solving for zeros of the Jacobian $\partial A(\mathbf{X})/\partial\mathbf{X}$ and results in a single path in model state space at the overall minimum of the action. The HMC method is a hybrid of these in which requires a symplectic integrator of the 'Hamiltonian' $H(\mathbf{P}, \mathbf{X}) = \mathbf{P}^2/2M + A(\mathbf{X})$





and uses $\partial A(\mathbf{X})/\partial \mathbf{X}$ to move about in 'canonical' $\{\mathbf{P}, \mathbf{X}\}$ space. Neither Monte Carlo method requires evaluating or storing higher derivatives of the action, and each samples the conditional probability distribution in path space, while the Laplace method does not. At this early stage of development of these methods, we do not have a firm recommendation as to which one to select in general. From the calculations on a high dimensional Lorenz96 model, it appears that on this test model, all approaches yield excellent results when enough measurements are made at each measurement time in an observation window.

# 6 Code Availability

All of the code needed to reproduce our results are available here.





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
