# Peer review of "Precision Annealing Monte Carlo Methods for Statistical Data Assimilation: Metropolis-Hastings Procedures"

_Nonlinear Processes in Geophysics, 2019_

## Referee Comment (RC1) · S.G. Penny (Referee) · 12 Mar 2019

General Comments:

Overall, this is a well presented introduction of the mathematical problem statement of data assimilation from the point of view of the authors, and the proposed DA solution method is interesting and worth pursuing. For this reason, I recommend publication after addressing a few items described in more detail below. Most importantly, I feel that more experiments can be performed to further illustrate the merits of the solution method, including comparison to other common solution methods and exploration of experiment parameter variations. Secondarily, it would be useful to provide some analysis of the cost of the algorithm, particularly indicating how man times the full nonlinear model has to be integrated at each stage.

—

The problem statement seems to lay out what would be considered a single "data assimilation cycle" - meaning that observations in one fixed time window are used to estimate the state and model parameters, and then produce a forecast beyond that time window. In forecasting applications, this process is cycled over multiple time windows, and that cycling has a dynamics of its own. The results section would benefit from testing some degree of this cycling process.

I'd like to see some basic analysis of the cost of the Monte Carlo annealing method. Most importantly, since in many realistic applications (e.g. in weather and climate prediction) the model is the most costly part, how many model integrations are needed at each step of the algorithm?

As a reader, it is difficult to evaluate the quality of the solution approach without one or more traditional methods to compare to as a baseline. Can the authors provide a comparison to analyses produced by a simple implementation of a 4D-Var and an EnKF method? Is the analysis and forecast accuracy similar? Are the computational costs significantly more or less expensive?

It should be straightforward to automate the generation of a more thorough set of experiment cases that would provide more robust validation of the solution method. I'd like to see some more sensitivity analysis based on varying the noise level in the observations, the number/distribution of observations used (for example, I expect far fewer observations may be needed if the locations are random at each analysis time), the size of the 'ensemble' used in the monte carlo approach, and the starting point using different points on the L96 attractor, for example.

A basic assessment of the Lyapunov spectrum for the L96 model with its given configuration (dimension and forcing) would help to give some context for the DA system performance. The presentation would benefit from a brief description of these at both the global (the maximum LE was referenced briefly) and local (i.e. FTLEs during the analysis and forecast time window) scales.

Specific Comments:

L18:

What is the difference in definition between data assimilation and statistical data assimilation?

L22-23:

Some may also have interest in the probability distribution or higher moments of the distribution.

L33:

It might be worth clarifying:

"...information from [measured] data..."

L 41:

We also want to estimate the measured states which are inaccurately known due to errors in the measurement devices.

L 45:

It might be worth mentioning that in prediction applications, this process is repeated by redefining the time window to start at t_F.

L 52:

Could the bold formatting vector notation from the text for functions F and f be used consistently in equations (1) and (2).

L 57:

Perhaps since 'F' is already used as a function name, another symbol could be used for the largest 'k' value index.

L 66:

It should maybe be mentioned that this assumes the model parameters are global, i.e. they are not time-dependent or spatially dependent. Though you could also point out that if you were to cycle the analysis window, without any other changes, it would support a slow time varying global forcing parameter as is.

L 91:

Is there a part of the process that is analogous to the validation set?

Assuming these definitions, for example:

"Training Set: this data set is used to adjust the weights on the neural network.

Validation Set: this data set is used to minimize overfitting by verifying that any increase in accuracy over the training data set also yields an increase in accuracy over a data set that has not been shown to the network before.

Testing Set: this data set is used only for testing the final solution in order to confirm the actual predictive power of the network."

Equation (3):

This may be an item for the editors to address, but the equation is difficult to read and would benefit from improved formatting.

I assume the large 'dot' is multiplication. It seems that the second large 'dot' (out of three) does not belong.

I like the use of color in the equation, but it might help to explain what it's purpose is.

L 101:

I think there's an error in the denominator, please revise.

L 105:

Perhaps it is simply semantics, but I interpret equation (4) as being built from propagating forward in time from time $t\_0$ to $t\_F$. Is there a reason for the interpretation of moving backwards in time?

Equation (6):

I'm assuming the summation with index "k" should be inside the left bracket, since the second term doesn't have any "k" indices. Perhaps also double check the placement of the negative signs once the terms are adjusted.

L 115:

Maybe I misinterpreted the notation, but it seemed based on equation (2) that $x(n+1)$ and $f(x(n),p)$ were equal. If this is supposed to represent a measure of model error, it would help if this was explained in more detail.

Equation (7):

Since the argument to the action A is cap X, and this doesn't appear in the right hand side, it is a bit confusing at first sight. Could the authors make the relationship more clear. Again, it would help if somewhere it is clarified how the terms $x(n+1)$ and $f(x(n),p)$ differ. Which is the known quantity and which is the control variable corresponding to X? I'd like a little more explanation of the quantities in the second term and where they come from.

L 132:

Maybe change "perform" to "solve" or "evaluate" the integral.

L 133:

"[occurs at] the [value of X that produces the] maxima of. . ."

L 138:

"methods to [solve] the integral. . ."

L 148:

change "yielding the smallest value of" to "minimizing"

L 170-172:

Could this be written more simply (or additionally) as an equation?

L 173:

"In practice, [the closer] one can choose X[init] to [correspond to] the maximum of P(X|Y), . . ."

L 176:

"[was] located (Shirman 2018), we [would]. . ."

L 184-186:

it should be acknowledged that this will encounter problems if the model has significant systemic errors relative to the 'true' system used to generate the observed data. (The systemic model errors may be more severe than a misspecified model parameter, e.g. a process that was oversimplified in the formation of the model).

I expect there should be some limitations depending on the number of observations that are used. For example, if L=1, then there is still a significant portion of the state estimate that will be unconstrained when Rf=0, and could lead to a quite poor starting condition. If there is not any discussion about this in the coming sections, I'd ask that some be added.

Equation (8):

This definition should probably appear where X first appears in the manuscript.

L 214:

There's an extra space before the period in "Eq. (9) ."

L 217-218:

"However, we substitute for $x_l(t_0 + k[n\tau])$, whenever it occurs in the equations Eq. (9), . . ."

I assume then that you're relying on a degree of synchronization so that the unobserved variables become dynamically consistent with the observed variables. With respect to this process, I have two questions:

(1) how long does it take for the unobserved variables to reach a dynamic equilibrium with the observed variables so the full states are balanced. Is this performed before the DA analysis window $[t\_0, t\_F]$, or within this window? If the latter, is sufficient time given in the ensemble generation phase to achieve sufficient 'spin up'?

(2) Assuming there is noise in the observed variables, this could potentially produce states that are not dynamically consistent. What is the sensitivity of this procedure to increasing noise in the observations?

L 227-229:

It would be helpful to explain the fixed iterations of M-H, what a 'burn-in' step is, etc. The ensemble generation stem is important and at the moment it seems a bit ambiguous, which makes analyzing the following steps more difficult as a reader.

L 231:

What does the second argument in $N\_A(q,0)$ stand for?

Equation (10):

This looks like an ensemble mean of the $N\_A$ accepted paths for each of the $N\_I$ initial

paths. From the description in lines 227-230, it's not clear to me how these paths are generated.

L 238:

"All these paths have zero action."

Just to make sure I understand - the paths all have zero action because they all intersect at the observed points, and at this point the Action term only includes the first term that applies a penalty as the state deviates from the observed values. Is this interpretation correct?

L 242-245:

Could you clarify how many integrations of the full nonlinear model are required at each iteration of increasing R_f values? Is it one per each of the N_I ensemble members?

L 249:

missing space: "X[0]for"

L 251:

Is there a guarantee that this procedure finds the true minimum and does not get stuck in one of the local minima?

L 246-264:

In this case, I think this is more explanation than is needed and this iterative processes can probably be consolidated by just defining the first step and the induction step.

Equation (14):

I wonder if there is something that can be learned from computing and examining the error covariance matrix produced by this resulting ensemble. It would be interesting to see how the error covariance matrix changes/converges over each of these iterations from 0 to beta.

L 282-283:

It seems like you will eventually reach a point of 'overfitting' a model with systematic errors (e.g. slightly incorrect fundamental equations, not just parameter errors). Is there a stopping criterion to avoid overfitting? Going back to my earlier question, I wonder if there is an analogous part of the algorithm to the 'validation' phase of the deep learning machine learning process that could help identify overfitting.

L 293:

What is the length of the time window? (i.e. what is $t\_0$ and $t\_F$?)

L 301:

I assume the choices of L=5 to 12 is only applied for the D=20 case. But it mentions above (L290) that there are cases with D=5, does this experiment case also have a similar parameter list?

Equation (16):

I asked before, but I'd like clarification about which 'x' terms in equation (16) are the control variables (i.e. the ones corresponding to X). Based on equation (8), I assume it is the $x\_l$ and $x\_a$ terms? But I know you are also trying to estimate p. My main point is that it is unclear and needs to be laid out in more detail when first introduced.

It's not really consequential, but the order of ($y\_l - x\_l$) changed from the previous action equation (7). It might be better to keep it consistent.

Figure (2):

I wonder if you can limit the ensemble size $N\_l$ equal to the number of positive (+neutral) Lyapunov exponents, similarly to the minimum required ensemble size for the EnKF. Have you tried reducing the ensemble size $N\_l$ and testing the sensitivity of convergence to this size?

For reference, see Bocquet and Carrassi 2017:

https://www.tandfonline.com/doi/full/10.1080/16000870.2017.1304504

I assume this figure illustrates that the system cannot be observed with as few as 5 observed model grid points per analysis time step. It might be worth clarifying the authors' interpretation in the figure caption.

Could you describe which are the observed variables, as in the Figure (3) caption?

Figure (3):

"This leads the action to become effectively equal to the action itself"

I don't understand this statement.

Figure (4):

It seems like most of the members are not finding the correct value of the forcing parameter. Am I interpreting this correctly?

Figure (5):

Are these results aggregating the cases in Figure (4)? They don't appear to correspond. Or is Figure (4) before convergence?

Figure (6):

dt = 5.0 for the L96 model likely has pretty nonlinear error growth. (1) An estimate of the error growth over this window (e.g. the FTLEs) could be useful to set the context for how you might expect errors to grow during the forecast period. (2) With a systematic model error this might experience even greater sensitivity. I wonder if the authors could attempt a similar experiment with a shorter time window (e.g. with more approximately linear error growth), but cycle the process over multiple time windows like a realistic forecasting application.

Equation (18):

Make the location of the precision terms (Rm, Rf) consistent, and apply consistent use of () versus [] brackets for each term.

L 423:

Put the actual url - a reader of a printed copy of the text will not be able to see the link.

---

## Referee Comment (RC2) · Anonymous Referee #2 · 28 Mar 2019

The authors appear to introduce a new version of data assimilation, referred to as statistical data assimilation, that is Monte Carlo based. I found the manuscript very difficult to follow, and that distracted from the potential of the work it is presenting.

The major problem with the manuscript is the presentation. It is to dense and very difficult to follow the flow of the mathematics at times, especially when it is included in the sentence. the first major change that has to occur if for the notation to conform to that of Ide (1997) so it is possible for the reader to compare to other DA systems, rather than translating what we think you are doing. You refer to at one point a diagonal precision matrix, but how is that related to the error covariance matrices of current DA

systems?

Another troublesome point is the fact that you mention the Laplace approach but then define this mathematically, or if you did it was obvious and this flows back into my previous comment.

My other comments are below:

The grammar is quite bad in places with tenses and pluralities incorrect too many times.

What is beta? you perform a whole set of analyses on this parameter but it is never defined, nor is it named correctly. Having just looked over the manuscript i find that beta is defined in the caption of figure 2, this need to be in the text when it is first introduce so the reader is prepared for this to understand the discussion and the figure itself.

Equation 4: You make no reference to the two previous papers that introduced this formulation of 4DVAR, van Leeuwen and Evensen (1996) and Fletcher (2010).

Sections 3.1 and 3.2 need to be better presented, either in the form of a flow chart figure to indicate the steps, or in a table

the figure caption for Figure 3 is too long.

It appears from what i could see that there are no real conclusions about this work, nor is there a discussion on how this could be extended to more complex systems.

I do believe that the manuscript has potential but it also need to address the following questions for it to be really be considered to be published.: 1) How does this approach compare relative to CPU time to current variational and ensemble based system? 2) How operationally viable is this approach? You need to address the feature or wall clock time for this approach.

---

## Author Comment (AC1) · 1 Apr 2019

Thank you for your suggestions about our paper.

**Bold text** are the referee's specific comments, and non-bold text are our responses.

**the first major change that has to occur if for the notation to conform to that of Ide (1997) so it is possible for the reader to compare to other DA systems, rather than translating what we think you are doing.**

Our notation has been used widely, and to our knowledge, conforms to general use. X are state variables and Y measurements.

**You refer to at one point a diagonal precision matrix, but how is that related to the error covariance matrices of current DA systems?**

Precision matrices are well known to be inverse covariance matrices for Gaussian distributions.

**What is beta? you perform a whole set of analyses on this parameter but it is never defined, nor is it named correctly. Having just looked over the manuscript i find that beta is defined in the caption of figure 2, this need to be in the text when it is first introduce so the reader is prepared for this to understand the discussion and the figure itself.**

If this were so, it would be quite unacceptable. Fortunately $\beta$ is defined, we thought clearly, on p. 10 near equation (13).

**Equation 4: You make no reference to the two previous papers that introduced this formulation of 4DVAR, van Leeuven and Evensen (1996) and Fletcher (2010).**

We think this goes back to work of Bennett around 2001. In any case, this is not a paper about 4Dvar, It is a paper about how to use Monte Carlo methods in performing expected value integrals such as Eq. (5). As such, the references to 4Dvar are limited, and references to Monte Carlo methods more extensive, e.g. Neal's paper from 2011. This is relevant to the subject of our paper.

**the figure caption for Figure 3 is too long.**

Actually we find it quite useful, and about the right length.

**The grammar is quite bad in places with tenses and pluralities incorrect too many times.**

We have read the paper over in detail several times. We do not find what the referee says here to be correct. It could be our flaw, of course, and we ask that some specific examples be identified so we may look at them.

**How does this approach compare relative to CPU time to current variational and ensemble based system?**

Thank you for the comment/suggestion. We have included an estimate of the computation times for the results shown, and further details will be shared as per Referee Penny's comments.

**How operationally viable is this approach? You need to address the feature or wall clock time for this approach**

This not an operational paper. This is a theoretical method that we introduce that has not been considered before: MCMC with precision annealing (see the references). Again, this is not 4DVar which only seeks the minimum of A(X) = -log[P(X)]. MCMC also samples in the neighborhood of those minima. 4DVar requires the derivatives of A(X), MCMC does not. That was the motivation for exploring it.

---

## Author Comment (AC2) · 19 Apr 2019

Thank you for your detailed review. We were, unfortunately, unable to respond to all your (11-pages worth of) comments before the end of the discussion period. All the comments will be addressed and incorporated into the upcoming draft.

[Figure]

**General Comments:**

We agree with your suggestion of the computational cost analysis. But we are concerned that the study of cycling dynamics, performing more experiments, and comparing against more traditional methods, would be too ambitious for our one paper.

**L22-23:**

Some may also have interest in the probability distribution or higher moments of the distribution.

G(X) is more general than just the (scalar) moments; it may even be a vector valued function. By a proper choice of G(X), say $(X - \mu)^n$, we can represent any $n^{th}$ moment to evaluate and much more.

**L33:**

It might be worth clarifying: "... information from [measured] data ..."

Good point; thanks.

**L 41:**

We also want to estimate the measured states which are inaccurately known due to errors in the measurement devices.

We will include the motivation for wanting the measure states as well, as per your suggestion.

**L 45:**

It might be worth mentioning that in prediction applications, this process is repeated by redefining the time window to start at t_F.

Though it might be true, this paper does not use cycling.

**L 52:**

Could the bold formatting vector notation from the text for functions F and f be used consistently in equations (1) and (2).

We believe that using the subscript may be clearer for equations (7) and (16), which are of utmost importance. Equations (1) and (2) have, as the input, the full $D$-dimensional state of the system.
**L 57:**

Perhaps since 'F' is already used as a function name, another symbol could be used for the largest 'k' value index.

We think that it would be clearer to have F represent final, and there should be little confusion that the dynamical system F. But we can replace $t_F$ in Line 57 with $t_{Final}$ for clarity perhaps?

**L 66:**

It should maybe be mentioned that this assumes the model parameters are global, i.e. they are not time-dependent or spatially dependent. Though you could also point out that if you were to cycle the analysis window, without any other changes, it would support a slow time varying global forcing parameter as is.

It has been previously stated multiple times, eg L44, L54.

**Equation (3):**

This may be an item for the editors to address, but the equation is difficult to read and would benefit from improved formatting. I assume the large 'dot' is multiplication. It seems that the second large 'dot' (out of three) does not belong. I like the use of color in the equation, but it might help to explain what it's purpose is.

We are not sure how to improve the formatting for clarity, and we welcome specific recommendations. Thank you for pointing out the mistake. The 'dot' is replaced with an 'x' and the out-of-place 'dot' has been removed. The color is to emphasize the recursion relation, and will be added as a comment.

**L 101:**

I think there's an error in the denominator, please revise.

Yes, thank you. One should be a and the other should be b.

**Equation (6):**

I'm assuming the summation with index "k" should be inside the left bracket, since the second term doesn't have any "k" indices. Perhaps also double check the placement of the negative signs once the terms are adjusted.

Seem like the notation is a little off here, thanks for catching it. It has been fixed.

**L 132:**

Maybe change "perform" to "solve" or "evaluate" the integral.

Okay, evaluate.

**L 133:**

"[occurs at] the [value of X that produces the] maxima of. . ."

Typo, thanks.

**L 138:**

"methods to [solve] the integral. . ."

Typo, thanks.

**L 148:**

change "yielding the smallest value of" to "minimizing"

Typo, thanks.

**L 173:**

"In practice, [the closer] one can choose X[init] to [correspond to] the maximum of P(X|Y), . . ."

Thank you.

**L 176:**

"[was] located (Shirman 2018), we [would]. . ."

Typo, thanks.

**Equation (8):**

This definition should probably appear where X first appears in the manuscript.

A similar form is on L67, but maybe it should be reiterated here.

**L 214:**

There's an extra space before the period in "Eq. (9) ."

Yes, thanks.

**L 227-229:**

It would be helpful to explain the fixed iterations of M-H, what a 'burn-in' step is, etc. The ensemble generation stem is important and at the moment it seems a bit ambiguous, which makes analyzing the following steps more difficult as a reader.

We will add some slightl explanation for readers who are not familiar with Monte Carlo methods. However, Metropolis-Hasting is a commonly and widely used algorithm, so we will probably reference a notable paper here. A full tutorial here will be too lengthy. There are many hyper-parameters to tweak, and this is dependent on available computational resources, problem statements, systems, etc.

**L 231:**

What does the second argument in $N_A(q, 0)$ stand for?

$N_A(q, \beta)$ means the $\beta^{th}$ annealing step, to be specific, $R_f = R_{f0}\alpha^\beta$. We should explain this clearer before hand, probably with a table, but was previously addressed in Eq. (13).

**L 238:**

"All these paths have zero action."
Just to make sure I understand - the paths all have zero action because they all intersect at the observed points, and at this point the Action term only includes the first term that applies a penalty as the state deviates from the observed values. Is this interpretation correct?

Yes, the interpretation is correct.

**L 249:**

missing space: "X[0]for"

Thanks.

**L 251:**

Is there a guarantee that this procedure finds the true minimum and does not get stuck in one of the local minima?

The Metropolis-Hastings algorithm guarantees a probability to climb uphill and thus will be likely to get out of local minima. The annealing heuristic has also been shown, empirically, to find a "superior minimum'. With better tuned hyper-parameters, there is a higher chance to arrive at the global minimum, but never guaranteed.

**L 246-264:**

In this case, I think this is more explanation than is needed and this iterative processes can probably be consolidated by just defining the first step and the induction step.

We think that it is an appropriate amount of explanation.

**Equation (14):**

I wonder if there is something that can be learned from computing and examining the error covariance matrix produced by this resulting ensemble. It would be interesting to see how the error covariance matrix changes/converges over each of these iterations from 0 to beta.

Indeed we think that it is interesting. I am not sure if we still have room in this paper to add it here, but we could try it and see if anything interesting happens.

**L 301:**

I assume the choices of L=5 to 12 is only applied for the D=20 case. But it mentions above (L290) that there are cases with D=5, does this experiment case also have a similar parameter list?

Yes, the parameters are the same. We should clarify.

**Equation (16):**

I asked before, but I'd like clarification about which 'x' terms in equation (16) are the control variables (i.e. the ones corresponding to X). Based on equation (8), I assume it is the $x_l$ and $x_a$ terms? But I know you are also trying to estimate p. My main point is that it is unclear and needs to be laid out in more detail when first introduced. It's not really consequential, but the order of $(y_l - x_l)$ changed from the previous action

equation (7). It might be better to keep it consistent.

X is defined previously in line 65, but we could remind the readers here by restating things for clarity. Fixed the changed order of x and y.

**Figure (3):**

"This leads the action to become effectively equal to the action itself" I don't understand this statement.

Typographical mistake. "...equal to the measurement term itself"

**Figure (4):**

It seems like most of the members are not finding the correct value of the forcing parameter. Am I interpreting this correctly?

We will likely remove figure (4).

**Figure (5):**

Are these results aggregating the cases in Figure (4)? They don't appear to correspond. Or is Figure (4) before convergence?

[Figure]

We will likely remove figure (4).

**L 423:**

Put the actual url - a reader of a printed copy of the text will not be able to see the link.

Good point, thanks.

---

## Author Comment (AC3) · 17 May 2019

**L18:**

What is the difference in definition between data assimilation and statistical data assimilation?

They are the same.
**L 105:**

Perhaps it is simply semantics, but I interpret equation (4) as being built from propagating forward in time from time t_0 to t_F. Is there a reason for the interpretation of moving backwards in time?

Forward or backward here are both equivalent in this formulation. Remember that time is not sequential in this formulation, but (in the language of graphs) is more like a vertex between two nodes on an un-directed graph.

**L 115:**

Maybe I misinterpreted the notation, but it seemed based on equation (2) that x(n+1)and f(x(n),p) were equal. If this is supposed to represent a measure of model error, it would help if this was explained in more detail.

If we are using the perfect state, then $x(n+1) = f(x(n), p)$. However, we propose that the **estimated** state evolve as $x(n+1) = f(x(n), p) + N(0, R_f^{-1})$, where N is the normal distribution. We will add this in the paper.

**Equation (7):**

Since the argument to the action A is cap X, and this doesn't appear in the right hand side, it is a bit confusing at first sight. Could the authors make the relationship more clear. Again, it would help if somewhere it is clarified how the terms x(n+1) and f(x(n),p) differ. Which is the known quantity and which is the control variable

corresponding to X? I'd like a little more explanation of the quantities in the second term and where they come from.

We have restated/reiterated **X** for clarity.

**L 184-186:**

it should be acknowledged that this will encounter problems if the model has significant systemic errors relative to the 'true' system used to generate the observed data. (The systemic model errors may be more severe than a mis-specified model parameter, e.g. a process that was oversimplified in the formation of the model).

We expect there should be some limitations depending on the number of observations that are used. For example, with D = 5 and L = 1, there is still a significant portion of the state estimate that will be unconstrained when $R_f = 0$, and could lead to a quite poor starting condition.

Also, if the artificial model is completely unusable, in the sense that it produces exponentially large errors, then this method will inevitably fail, as would any method. But as long as the error are bounded in the $L_2$ sense, then the method should work. That is not to say that it will predict well, or that the results will look appealing, but it is the best one could do with the given (faulty) model and measurements.

**L 217-218:**

"However, we substitute for $x_l(t_0 + k[n\tau])$, whenever it occurs in the equations Eq. (9), ..." I assume then that you're relying on a degree of synchronization so that the unobserved variables become dynamically consistent with the observed variables. With respect to this process, I have two questions: (1) how long does it take for the unobserved variables to reach a dynamic equilibrium with the observed variables so the full states are balanced. Is this performed before the DA analysis window $[t_0, t_F]$, or within this window? If the latter, is sufficient time given in the ensemble generation phase to achieve sufficient 'spin up'? (2) Assuming there is noise in the observed variables, this could potentially produce states that are not dynamically consistent. What is the sensitivity of this procedure to increasing noise in the observations?

1) We don't know how much time it takes for the synchronization to occur, but this is missing the point. We just generate these paths within the window $[t_0, t_F]$, and they may or may not be 'good' relative to the true solution, but almost always 'good' compared to a random initial guess. We are simply trying to be a bit better than random. 2) This is a very interesting question, and one that we are aware of and will look into deeper. We have tried adding a generous amount of noise. So far, this method seems pretty resilient to noise. This method of initialization will be tackled in much greater detail in a separate paper.

**Equation (10):**

This looks like an ensemble mean of the $N_A$ accepted paths for each of the $N_I$ initial paths. From the description in lines 227-230, it's not clear to me how these paths are generated.

We use standard Metropolis-Hastings Monte Carlo sampling to generate the $N_A$ paths. We will defer the full M-H algorithm to some good review papers.

**L 242-245:**

Could you clarify how many integrations of the full nonlinear model are required at each iteration of increasing R_f values? Is it one per each of the N_I ensemble members?

There are a total of $M^2$ full model integration steps.

**L 282-283:**

It seems like you will eventually reach a point of 'overfitting' a model with systematic errors (e.g. slightly incorrect fundamental equations, not just parameter errors). Is there a stopping criterion to avoid overfitting? Going back to my earlier question, I wonder if there is an analogous part of the algorithm to the 'validation' phase of the deep learning machine learning process that could help identify overfitting.

In fact, the point of the algorithm is to "over-fit" the model to some extent. However, this is not quite the same as as overfitting or memorizing data. Moreover, if the fundamental equations are wrong, then most methods we are aware of will fail too. We can always stop the algorithm as soon as $\frac{dA}{dR_f} = 0$, so that $R_f$ does not get too large. From our experiments, we have not seen evidence/consequence for anything resembling as

overfitting. Hopefully, this addressed the concern of overfitting.

In the machine learning language, the estimation window is equivalent to the test set. And it seems, conceptually, that the prediction window is the equivalent of the validation set. As far as we can infer, there is no test set equivalent, nor is there a good reason for one.

**L 293:**

What is the length of the time window? (i.e. what is $t_0$ and $t_F$?)

t_0 is 0, and t_F is 5, given $\Delta t = 0.025$. (We should be adding this to the manuscript somewhere).

**Figure (2):**

I wonder if you can limit the ensemble size N_I equal to the number of positive (+neutral) Lyapunov exponents, similarly to the minimum required ensemble size for the EnKF. Have you tried reducing the ensemble size N_I and testing the sensitivity of convergence to this size? For reference, see Bocquet and Carrassi 2017: https://www.tandfonline.com/doi/full/10.1080/16000870.2017.1304504 I assume this figure illustrates that the system cannot be observed with as few as 5 observed model grid points per analysis time step. It might be worth clarifying the authors' interpretation in the figure caption. Could you describe which are the observed variables, as in the Figure (3) caption?

We have not tried anything like what you have referenced. We don't know the answer to this question, since we don't know if this is comparable to the ensemble size for the EnKF.

**Figure (6):**

dt = 5.0 for the L96 model likely has pretty nonlinear error growth. (1) An estimate of the error growth over this window (e.g. the FTLEs) could be useful to set the context for how you might expect errors to grow during the forecast period. (2) With a systematic model error this might experience even greater sensitivity. I wonder if the authors could attempt a similar experiment with a shorter time window (e.g. with more approximately linear error growth), but cycle the process over multiple time windows like a realistic forecasting application.

We do not quite understand this comment and are unable to respond to it, at this time, without further clarification.